# Relationships Among Origin, Genotype, and Oenological Traits of *Brettanomyces* Yeasts

**DOI:** 10.3390/ijms252111781

**Published:** 2024-11-02

**Authors:** Laura Canonico, Alice Agarbati, Francesca Comitini, Maurizio Ciani

**Affiliations:** Department of Life and Environmental Sciences, Polytechnic University of Marche, Via Brecce Bianche, 60131 Ancona, Italy; l.canonico@univpm.it (L.C.); a.agarbati@univpm.it (A.A.); f.comitini@univpm.it (F.C.)

**Keywords:** *Brettanomyces*, cluster analysis, biotyping, winemaking environments, fermentation characters

## Abstract

*Brettanomyces* yeasts play a relevant role in the fermentation industry, showing controversial behavior. There is growing interest in these yeasts in the fermentation industry as beer and bioethanol production, while in winemaking, they are considered spoilage microorganisms mainly used to produce ethyl phenols. These compounds may alter wine’s organoleptic characteristics, leading to significant economic loss. In this work, 45 *Brettanomyces* strains from seven different environments were genotyped and assayed for some oenological characters to investigate the possible relationship among sources of isolation, genotype characterization, and oenological characters. The results of biotyping showed four main clusters which were also distinguished according to the oenological characters. The oenological characters also distinguished the strains based on the isolation source, suggesting an overall relation between origin and biotypes. The negative correlation between fermentation rate and ethyl phenols production in the *Brettanomyces* population may indicate the adaptation to hostile environments differently from crop strains that showed the opposite behavior. The overall results contribute to clarifying some features of *Brettanomyces* yeasts, even if further investigations into the ability of these yeasts to colonize winemaking environments are needed.

## 1. Introduction

*Brettanomyces* yeasts have been isolated in a wide diversity of environments and substrates. They are hardly isolated from natural substrates such as plants or fruit surfaces, while they are more easily isolated from fermented food and beverages [1,2]. *Brettanomyces* yeasts have acquired growing interest in the biotechnological industry, particularly in beer, wine, and ethanol production. These yeasts are facultative anaerobic and Crabtree-positive and produce high amounts of acetic acid and ethanol under aerobic conditions [3]. In some substrates and environments, *Brettanomyces* yeasts may play a positive role due to their ability to compete with yeast microbiota in specific conditions such as during the production of Lambic beer with the production of pleasant flavors [4].

In winemaking environments, *Brettanomyces* yeasts are considered spoilage microorganisms that alter the organoleptic characteristics of the wine, leading to significant economic losses throughout the world [5]. Indeed, *Brettanomyces bruxellensis*, also known as the teleomorph *Dekkera bruxellensis*, is the main species recognized in the winemaking environment that confers unwanted olfactory characteristics defined as “horse sweat” taint in the bulk or bottled red wines (“Brett characters”) [6] through the production of volatile off-flavors, mainly ethyl phenols [7]. In the last two decades, a wide range of papers have investigated the *B. bruxellensis* species and its involvement in winemaking. Most of these studies focused on the production of volatile phenols, as responsible for defects [8,9], on the biotic and abiotic factors that influence the growth of *B. bruxellensis* [10,11,12], or some peculiarities of the species, such as the ability to enter the VBNC (Viable but Non Culturable) form [13].

Other studies have been carried out to investigate the adaptation and dominance of *Brettanomyces* yeasts in different adverse environments such as wine [2,8,14]. The strong genetic diversity encountered in wine strains increases the chances of adaptation to all types of wine. Several investigations on *B. bruxellensis* have been published [15,16,17,18,19,20], displaying a wide intraspecific variability [7,15,21].

Nevertheless, not all strains may present the same wine spoilage potential. A series of works conducted in terms of carbon metabolism, stress tolerance, or production of volatile phenols revealed significant differences between strains [2,8,11,22,23,24,25,26,27]. However, the relationship between genetic variability, origin, and oenological characters showed variable and contradictory results.

In the present work, 42 strains of *B. bruxellensis* and 3 *Brettanomyces anomalus* strains from different habitats such as wine, beer, spontaneous process kombucha, and crops (cocoa, corrosol) were genotyped and tested for the oenological characters of interest. The aim was to investigate the relationship among sources of isolation, genotype, and phenotypic traits of oenological interest.

## 2. Results

### 2.1. Biotyping of Brettanomyces spp.

The 45 strains of *Brettanomyces* yeasts under evaluation belonging to the DiSVA Collection were submitted to the typing process using the combination of five different primers. The results showed four main clusters, as reported in the constellation plot using Ward clustering (Figure 1).

The first main abundant cluster (cluster 1, red dot) was constituted by the biotypes from grape and winery environments from the winemaking area denominated Tuscany 1, apart from the biotype wF9a that was enclosed in cluster 3 (green dot) together with the biotypes from Cameroon (cocoa seeds) and belonging to B. anomalus species. Cluster 2 (orange dot) was composed of only two biotypes: biotype C2.3 *B. bruxellensis* from Cameroon (corrosol) and biotype M6 (winery Marche region). The broad cluster 4 (blue dot) included two well-distinct sub-clusters: (i) biotypes from the winemaking area Tuscany 2; (ii) biotypes from the winemaking area Marche region and Lombardy region, as well as beer biotypes. Apart from some exceptions, the biotypes were well distinguished in clusters and sub-clusters on the basis of the environmental habitats of isolation.

### 2.2. Analysis of Oenological Parameters

The results of the evaluation of the influence of oenological characters on the bases of the clusters obtained from the biotype analysis are shown in Table 1.

The results of the ethanol production showed a limited differentiation among the clusters (only cluster 3 showed high values), while cluster 1 can be distinguished for the low fermentation rate values exhibited. Ethyl phenol production did not show differences among the clusters, while the production of ethyl guaiacol showed significant differences only between clusters 1 and 2. Finally, volatile acidity distinguished clusters 1 and 2 (lower amounts) from clusters 3 and 4 (higher amounts). The oenological traits, evaluated by microfermentation trials using commercial grape juice showed a wide variability at the strain level (Appendix A). The results of the oenological characters were then evaluated in terms of the function of the environmental isolation of the strains (Table 2).

The ethanol production (% *v*/*v*) did not show significant differences among the strains grouped based on the source of isolation. Only Tuscany 2 and Gueze beer groups could be statistically separate (high and low ethanol production, respectively). The parameter of the fermentation rate showed more variability among the strains grouped for different origins. Tuscany Winery 1 and Lombardy Winery showed a significant low fermentation rate, while Marche winery and the Cameroon strains exhibited a significant high fermentation rate. Other parameters evaluated, such as volatile acidity, ethyl phenol, and ethyl guaiacol production, did not show differences, apart from the Gueze beer group that exhibited significantly high ethyl phenols.

### 2.3. Relationship Among Oenological Characters and Multiparametric Evaluations

The evaluation of the correlation among the different oenological characters in the whole population of *Brettanomyces* strains highlighted a significant negative correlation between fermentation rate and both ethyl phenol compounds (ethyl guaiacol and ethyl phenol) (Figure 2). No further correlations in the other oenological characters were found.

To evaluate the relationship between source of isolation, biotype, and the oenological traits of *Brettanomyces* yeasts, a multiparameter approach, as a principal component analysis (PCA), was used (Figure 3).

The elaboration of the results in the function of the clusters from biotypes and reported in Figure 3a showed a clear distinction of the clusters in the four quadrants with the explanation of the total variability of 80% (PCA1 55%; PCA2 25%). Cluster 3 was characterized by ethanol production, while cluster 1 was influenced by the production of ethyl phenols. Cluster 2 in the left upper quadrant and cluster 4 in the left low quadrant were more influenced by fermentation rate and to a lesser extent by volatile acidity. The same factors were elaborated on in the function of the groups formed on the bases of the origin of *Brettanomyces* yeasts (Figure 3b). The distribution of the factors and the total variance explained (80.6%) were very close to that detected in Figure 3a (clusters’ distribution). The distribution of *Brettanomyces* yeasts showed a general distribution in the function of the environment of isolation. Indeed, in the upper quadrants, the largest groups of winemaking origin were placed (Tuscany 1 and Tuscany 2 right quadrant and Marche left quadrant). Biotypes from Gueze beer were placed in the right down quadrant, while the other groups were placed in the last quadrant. These results suggested a high similarity between biotypes and environment of isolation of the *Brettanomyces* strains. Moreover, the same results showed that the main oenological traits clearly distinguished the *Brettanomyces* yeasts population on the bases of the results of biotypes and their source of isolation, indicating a related relation between biotypes and origin of *Brettanomyces* yeasts.

## 3. Discussion

The yeast *Brettanomyces* is a good example of an organism facing human-driven selection pressures, where the broad genetic and phenotypic diversity makes its study extremely important to overcome their spoilage effect. Population genomic studies have revealed the coexistence of auto- and allotriploidization events with different evolutionary outcomes [28]. A wide investigation of the genotypic survey of *B. bruxellensis* using microsatellite analysis found that the population is structured according to ploidy level, substrate of isolation, and geographical origin of the strains [29]. They found that clustering was correlated to variable stress response, and the geographical origin has a different contribution to the population structure, strongly suggesting an anthropic influence on the spatial biodiversity of this microorganism. In this study, these results were confirmed. Indeed, the analysis of the *Brettanomyces* population also using a multiparametric evaluation showed the relation between the origin of isolation and genotype profiles of strains with the most relevant oenological traits. However, specific phenotypic traits are insufficiently investigated considering the great genomic diversity of these yeast species. The influence in these relations is different among these parameters. Some oenological parameters, such as theoretical ethanol and volatile acidity, played a minor role in the discrimination of the biotypes, while the parameters fermentation rate (Tuscany1, cluster 1) and ethyl phenols production (Guese beer, cluster 1 and 2) played a greater role in the relationship with origin and phenotype groups. In this regard, we found that fermentation rate and ethyl phenols production showed a significant and negative correlation that could be of interest to understand the adaptation of these yeasts in hostile environments where the dominance of the ecological niche is related to the production of ethyl phenols. The production of volatile phenols by *B. bruxellensis* strains is variable and has been extensively described in oenological conditions [8,23,25,27,30,31,32,33]. Most studies involved end-point analyses, which makes it difficult to distinguish between the intrinsic capacities of the strains and their modulation by external parameters. Furthermore, the respective importance of the different factors governing such variability is poorly described; a few authors showed that volatile phenols’ production was both strain-dependent and wine matrix-dependent [34,35,36,37]. The abiotic characteristics of isolation environmental habitat could also play a role as well as brewing strains, characterized by a more efficient metabolism toward ferulic acid (leading to 4-EG) than p-coumaric acid compared to wine isolates [38]. Some wine and soft drink isolates possess a duplication of the vinyl phenol reductase gene, which is absent in beer isolates [24]. However, while all these studies confirmed the existence of intraspecific variation, their impact on the production of volatile phenols and their possible ecological significance remains to be properly assessed.

The conversion of hydroxycinnamic acids (HCAs) into volatile phenols may have an important impact from an ecological viewpoint; HCAs have antimicrobial properties, and the ability to convert HACs into less toxic compounds could promote yeast growth [39]. The significant and negative correlation between fermentation rate and ethyl phenols production may confirm this role. Indeed, some environmental *Brettanomyces* strains (Cameroon strains) showed high fermentation rates and low ethyl phenols production. However, further data and insights are needed to clarify this behavior.

Regarding the genetic variability, a high genetic diversity of *B. bruxellensis* isolates from wine environments was found [35]. On the other hand, each specific genotype could be isolated repeatedly in the same winery over decades, demonstrating an unsuspected persistence ability. Besides stability in the winery, a great geographic dispersal was also evidenced, with some genotypes isolated in wines from different continents. The results obtained here clearly displayed high variability among different groups clustered based on genotype and origin. The results of the principal component analysis (PCA) clearly showed a strict correlation between the geographical and isolation origin of strains and the oenological traits of *Brettanomyces* strains.

Although ecology studies aiming to explain the origin of wine yeasts have mainly been carried out on *Saccharomyces cerevisiae*, as much research has been published studying *Brettanomyces*/*Dekkera* strains. Over fifteen years ago, [40] reviewed the latest knowledge concerning the role of *B. bruxellensis* in red wine to establish their origin and alterative role. The already established low sensitivity of *B. bruxellensis* to osmotic pressure and SO_2_ [41] as well as ethanol concentration [42] mean that the grape must have an environment which is more favorable to its growth; fermentations are described as key steps in *Brettanomyces* development management. The concentration of phenolic acids in general, and hydroxycinnamic acids in particular, could play an important role in volatile phenol production, suggesting that *B. bruxellensis* efficient survival in such challenging conditions is due to mechanisms unique to it, together with a conserved yeast stress response.

## 4. Materials and Methods

### 4.1. Yeast Strains and Media

The 45 yeast strains used in this study belong to *B. bruxellensis* and *B. anomalus* species and were isolated from four different Italian wine-making environments, Gueze beers, and fruit/seed surfaces from Cameroon; these are reported in Table 3 [43,44]. Strains DBVPG 6706 (type strain) and 6710 (Department of Plant Biology of the University of Perugia) isolated from Lambic beer and kombucha samples, respectively, were also enclosed. Pure isolates were maintained at 4 °C on YPD medium (yeast extract 1% *w*/*v*, peptone 2% *w*/*v*, dextrose 2% *w*/*v*, and agar 2% *w*/*v*) (Oxoid, Hampshire, UK) for short-term storage. They were stored at a temperature of −80 °C in a cryopreservation medium containing glycerol (40%) for long-term storage.

### 4.2. Molecular Identification and Biotyping

*Brettanomyces* strains were identified by molecular analysis through sequence analysis of the 26S-D1/D2 rDNA region, which identified species-specific nucleotide sequences [45].

The DNA extracted from the *B. bruxellensis* and *B. anomalus* strains underwent biotyping using the RAPD-PCR primers M13 (5′-GAG GGT GGCGGT TCT-3′); M14 (5′-GAGGGT GGG GCC GTT-3′); OPC20 (5′-ACTTCGCCAC-3′); and OPK03 (5′-CCAGCTTAGG-3′) and the minisatellite primers PIR1 (forward: 5′-GCCACTACTGCTTCCTCCAA-3′; reverse: 5′-TG GACCAACCAGCAGCATAG-3′) and PIR3 (forward: 5′-TCCT CCGTCGCCTCATCTAA-3′; reverse: 5′-GGCACTGAGAACCA ATGTGC-3′) [43,46]. For molecular typing, the strains from the University of Perugia collection DBVPG 6706 and 6710 were also included. The conditions for the amplification protocols were the ones described by Crauwels et al. [16] and Canonico et al. [46] (reported in Appendix A). Different genetic profiles using the RAPD-PCR and minisatellite primers described above define the biotype (strain). Samples of 15 μL of PCR products were loaded onto 2.5% (*w*/*v*) agarose gels, and the electrophoresis was carried out at 70 V for 2 h in 0.5× TBE buffer. The DNA bands on the gel were visualized by staining with ethidium bromide, and acquisition of the images was performed under a UV lamp (UV source GelDoc 1000; Bio-Rad, Milan, Italy). The length of the PCR products was estimated by comparing them with a 100 bp marker DNA standards (GeneRuler 100 bp DNA Ladder; AB Fermentas Inc., Waltham, MD, USA).

### 4.3. Microfermentation Trials

The fermentation traits were evaluated by microfermentation trials.

The trials were carried out on 70 mL of pasteurized commercial grape juice (Folicello Modena, Italy) from Lambrusco Grasparossa of Castelvetro and Sangiovese grapes (vintage 2016) that have the following initial composition: 19.1% reducing sugars, nitrogen content YAN 115 mgN/L, and pH 3.4.

The pasteurization process was carried out at 100 °C for 10 min. Modified YPD (0.5% *w*/*v* yeast extract, 2% *w*/*v* glucose, 0.1% *w*/*v* peptone) (Oxoid) was used to pre-culture all strains for 5–7 days at 25 °C in an orbital shaker (rotation, 150 rpm) (Thermo Fisher Scientific Inc., Milan, Italy). The biomass was used to inoculate the grape juice at the initial concentration of approximately 1× 10^6^ cells/mL for each strain. The cell concentration was determined using a Thoma–Zeiss counting chamber (Merck KGaA, Darmstadt, Germany). The fermentation trials were conducted at 25 °C ± 2°C under static conditions in duplicate.

### 4.4. Fermentation Parameters and Analytical Compounds

The weight loss of the flasks due to the CO_2_ evolution until the end of fermentation (i.e., constant weight for 2 consecutive days) was used to determine the fermentation kinetics. The volatile acidity was measured using the current analytical methods according to the Official European Union Methods [47].

The 4-ethyl guaiacol and 4-ethyl phenol compounds were quantified using the solid-phase microextraction (HS-SPME) method as described by Canonico et al. [48]. The fiber Divinylbenzene/Carboxen/Polydimethylsiloxane (DVB/CAR/PDMS) (Sigma-Aldrich, St. Louis, MO, USA), after adsorbed procedures, were desorbed into a GC injector (GC-2014, Shimadzu, Kyoto, Japan) equipped with an FID detector using a glass Supelcowax-10 column (60 m × 0.32 mm × 0.25 mm) from Supelco^®^ (Sigma-Aldrich). Helium was used as the carrier gas (flow rate of 3.74 mL/min). The fiber was inserted on the GC-port (230 °C) for 5 min to transfer the analytes to the chromatographic column. A split/splitless injector was used in the splitless mode: 60 s splitless; injection and detector temperature, 230 °C. The column program was 50 °C for 1 min and increased 2 °C/min to 200 °C and maintained at 200 °C for 20 min. Calibration curves set up for each compound were used to identify and quantify the compounds analyzed. 3-octanol was used as the internal standard (Sigma-Aldrich) at a concentration of 1.6 mg/L.

### 4.5. Statistical Analysis

The genotypic analysis of the strains into biotype clusters was performed using JMP version 11 ^®^ (statistical discovery from SAS, New York, NY, USA). The cluster analysis was carried out according to the Ward’s minimum variance method, and it was represented as a constellation plot. This diagram arranges the samples as endpoints and each cluster joins as a new point. The lines represent membership in a cluster. The length of a line between cluster joins approximates the distances between the clusters that were joined. The main analytical characters of clusters have been subjected to analysis of variance (ANOVA). The significant differences among the clusters were determined using Duncan’s test, and the results were considered significant if the associated *p*-values were <0.05. The analytical characters were also evaluated by using the bivariate estimate of characters and the principal component analysis (PCA) to determine the eventual correlation among the characters and the discrimination between the means and the variability among the strains tested. Statistical software package JMP 11^®^ (SAS, Cary, NC, USA) was used for the PCA statistical analysis.

## 5. Conclusions

The adaptation of *B. bruxellensis* to winemaking conditions involves several different factors related to stressful environments and anthropic factors. Here, it was found that oenological traits of interest were able to distinguish the *Brettanomyces* yeast population in the function of genotype and origin, also suggesting a relationship between them. The geographical origin contributes to the population structure. This result strongly suggests an anthropic influence on the spatial biodiversity of this microorganism. The negative correlation found between the fermentation rate and the production of ethyl phenols suggested that the efficient metabolism of some strains of *Brettanomyces* yeasts toward cinnamic acids could be associated with low fermentation rate and ultimately to the ability to colonize difficult environments, while crop strains (Cameroon strains) showed the opposite behavior. Further investigations into specific phenotypic traits are needed, as well as the ability to colonize harsh environments considering the great genomic diversity of these yeast species.

## Figures and Tables

**Figure 1 ijms-25-11781-f001:**
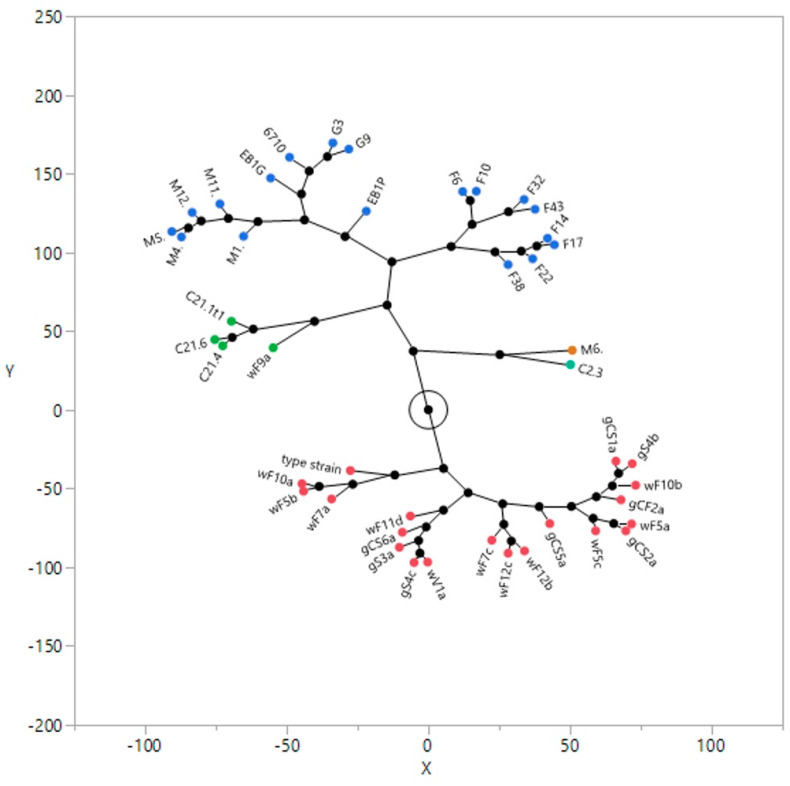
Ward clustering and constellation plot of the *Brettanomyces bruxellensis* biotypes using M13, M14, OP C20, and OPK03 RAPD primers and the PIR1 and PIR3 minisatellite primers. The representation of the clusters is 
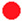
 cluster 1; 
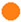
 cluster 2; 
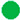
 cluster 3; and 
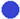
 cluster 4.

**Figure 2 ijms-25-11781-f002:**
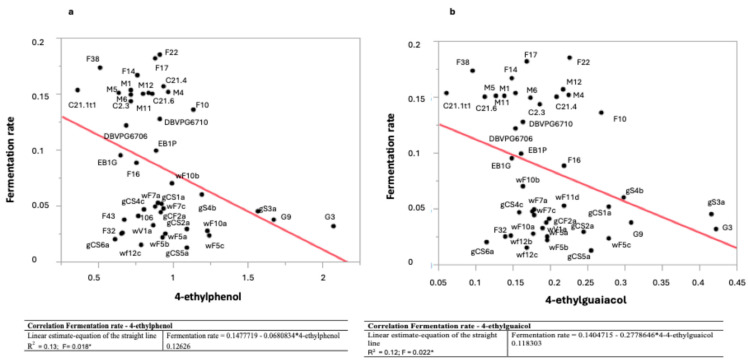
Correlation in the population of *Brettamomyces* strains between fermentation rate and ethyl phenol (**a**) and ethyl guaiacol (**b**). * Significant at *p* < 0.05.

**Figure 3 ijms-25-11781-f003:**
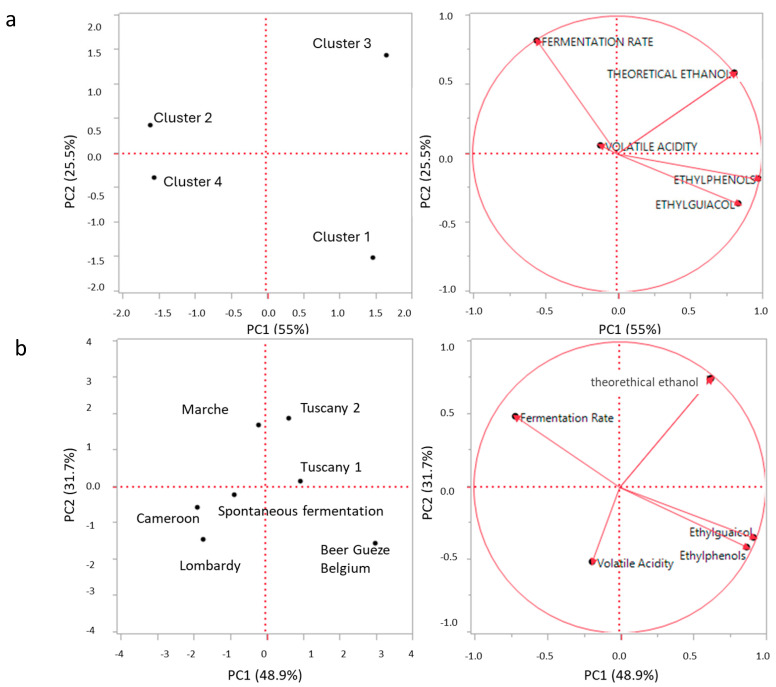
Principal component analysis (PCA) of oenological characters from different clusters of *Brettanomyces* yeast strains’ fermentation. The variance explained by PCA is PC 1 = 55% x-axis and PC 2 = 25.5% y-axis (**a**); principal component analysis (PCA) of oenological characters in function of the groups formed on the bases of the origin of *Brettanomyces* yeasts. The variance explained by PCA is PC 1 = 48.9% x-axis and PC 2 = 31.7% y-axis (**b**).

**Table 1 ijms-25-11781-t001:** The data of the principal fermentation characters are elaborated within each cluster for the analysis of variance (ANOVA). Data are means ± standard deviations, and those with different superscript letters (^a–c^) within each column are significantly different according to Duncan’s test (*p*-value < 0.05).

CLUSTER	Fermentation Rate gCO_2_/Day *	Theoretical Ethanol (% *v*/*v*)	Volatile Acidity (mg/L)	4-ethyl Phenol (mg/L)	4-ethyl Guaiacol (mg/L)
Cluster 1	0.042 ± 0.025 ^b^	8.59 ± 2.06 ^ab^	1.214 ± 0.591 ^c^	0.957 ± 0.251 ^a^	0.208 ± 0.075 ^a^
Cluster 2	0.126 ± 0.053 ^a^	8.38 ± 1.49 ^ab^	0.898 ± 0.349 ^c^	0.727 ± 0.255 ^a^	0.146 ± 0.070 ^b^
Cluster 3	0.120 ± 0.056 ^a^	9.30 ± 2.13 ^a^	1.436 ± 0.697 ^b^	0.907 ± 0.435 ^a^	0.194 ± 0.086 ^ab^
Cluster 4	0.116 ± 0.00 ^a^	8.17 ± 0.00 ^b^	1.865 ± 0.000 ^a^	0.713 ± 0.00 ^a^	0.178 ± 0.000 ^ab^

* Fermentation rate calculated over the 6th day of fermentation.

**Table 2 ijms-25-11781-t002:** The data of the principal fermentation characters are elaborated within each environment to analysis of variance (ANOVA). Data are means ± standard deviations, and those with different superscript letters (^a–c^) within each column are significantly different according to Duncan’s test (*p*-value < 0.05).

Environment	Fermentation Rate gCO_2_/Day *	Theoretical Ethanol (% *v*/*v*)	Volatile Acidity (mg/L)	4-ethyl Phenol (mg/L)	4-ethyl Guaiacol (mg/L)
Vicarello (Tuscany 1, Italy)	0.039 ± 0.015 ^c^	8.68 ± 1.90 ^ab^	1.02 ± 0.52 ^a^	0.960 ± 0.220 ^b^	0.206 ± 0.073 ^b^
Spontaneous fermentation (DBVPG)	0.125 ± 0.005 ^ab^	8.56 ± 0.76 ^ab^	1.88 ± 0.29 ^a^	0.740 ± 0.230 ^b^	0.157 ± 0.007 ^b^
CAMEROON	0.151 ± 0.006 ^a^	7.52 ± 0.93 ^ab^	1.32 ± 0.69 ^a^	0.727 ± 0.255 ^b^	0.143 ± 0.071 ^b^
Winery (Italy, Marche region)	0.151 ± 0.001 ^a^	9.25 ± 0.46 ^ab^	1.07 ± 0.23 ^a^	0.781 ± 0.106 ^b^	0.169 ± 0.039 ^b^
Red wine (Italy, Lombardy)	0.097 ± 0.003 ^b^	7.13 ± 0.35 ^b^	1.63 ± 0.46 ^a^	0.762 ± 0.167 ^b^	0.153 ± 0.009 ^b^
Gueze Beer (Belgium)	0.035 ± 0.005 ^c^	8.58 ± 1.59 ^ab^	1.59 ± 0.84 ^a^	2.170 ± 0.069 ^a^	0.364 ± 0.081 ^a^
Winery (Italy, Tuscany 2, Chianti)	0.125 ± 0.066 ^ab^	10.11 ± 1.772 ^a^	1.53 ± 0.89 ^a^	0.787 ± 0.199 ^b^	0.187 ± 0.046 ^b^

* Fermentation rate calculated over the 6th day of fermentation.

**Table 3 ijms-25-11781-t003:** List of 45 *Brettanomyces* strains used in this study.

Strain Code	Specie	Source of Isolation
M1	*B. bruxellensis*	Winery (Italy, Marche region)
M4	*B. bruxellensis*	
M5	*B. bruxellensis*	
M6	*B. bruxellensis*	
M11	*B. bruxellensis*	
M12	*B. bruxellensis*	
C2.3	*B. bruxellensis*	Cameroon (isolated from Corrosol)
C 21.4	*B. anomalus*	Cameroon (isolated from Cocoa seeds)
C 21.6	*B. anomalus*	
C 21.1 t1	*B. anomalus*	
EB1G	*B. bruxellensis*	Red wine (Italy, Lombardy)
EB1P	*B. bruxellensis*	
F6	*B. bruxellensis*	Winery (Italy, Tuscany 2, Chianti)
F10	*B. bruxellensis*	
F14	*B. bruxellensis*	
F17	*B. bruxellensis*	
F22	*B. bruxellensis*	
F32	*B. bruxellensis*	
F38	*B. bruxellensis*	
F42	*B. bruxellensis*	
G3	*B. bruxellensis*	Gueze beer (Belgium)
G9	*B. bruxellensis*	
DBVPG 6710	*B. bruxellensis*	Spontaneous fermentation (DBVPG)
DBVPG 6706	*B. bruxellensis*	
gCS1a	*B. bruxellensis*	(Tuscany 1, Italy) grape surface and winery
gCS2a	*B. bruxellensis*	
gCS5a	*B. bruxellensis*	
gCS6a	*B. bruxellensis*	
gS4b	*B. bruxellensis*	
gS4c	*B. bruxellensis*	
gS3a	*B. bruxellensis*	
gCF2a	*B. bruxellensis*	
wF5a	*B. bruxellensis*	
wF5b	*B. bruxellensis*	
wF5c	*B. bruxellensis*	
wF7a	*B. bruxellensis*	
wF7c	*B. bruxellensis*	
wF9a	*B. bruxellensis*	
wF10a	*B. bruxellensis*	
wF10b	*B. bruxellensis*	
wF11d	*B. bruxellensis*	
wF12b	*B. bruxellensis*	
wF12c	*B. bruxellensis*	
wV1a	*B. bruxellensis*	

## Data Availability

The original contributions presented in this study are included in the article/Appendix A; further inquiries can be directed to the corresponding author.

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
