# Peer review of "Relationships Among Origin, Genotype, and Oenological Traits of Brettanomyces Yeasts"

_ijms, 2024, doi:10.3390/ijms252111781_

Round 1

Reviewer 1 Report

Comments and Suggestions for Authors

A very interesting manuscript on the biotyping and oenological evaluation of several Brettanomyces strains of different origin. The manuscript is very well prepared, the materials and methods are suitable for such a study, the results are clearly presented and adequately discussed. Only a couple of minor improvements can be suggested:

Materials and Methods

Paragraph 4.2. Please add the PCR and electrophoresis conditions, it will be very helpful for the reader who wants to perform similar experiments.

Paragraph 4.5. Please explain the process from genotypic profile to biotype clusters.

Results

Please add representative photographs of genotypic profiles for each of RAPD-PCR primers.

Please pay attention to the use of italics; genera should be written in italics (e.g. l. 128, 130, 139-149 and possibly elsewhere), species should be written with the first letter in lowercase (e.g. l. 333, 336 and possibly elsewhere)

Author Response

A very interesting manuscript on the biotyping and oenological evaluation of several Brettanomyces strains of different origin. The manuscript is very well prepared, the materials and methods are suitable for such a study, the results are clearly presented and adequately discussed. Only a couple of minor improvements can be suggested:

Materials and Methods

Paragraph 4.2. Please add the PCR and electrophoresis conditions, it will be very helpful for the reader who wants to perform similar experiments.

Response :  we added in the text the electrophoresis conditions while pcr conditions were enclosed in the representative photographs of genotypic profiles for each of RAPD-PCR primers in supplemental materials

Paragraph 4.5. Please explain the process from genotypic profile to biotype clusters.

Response :We prepare a binary matrix comparing and summing the electrophoretic profiles obtained from each primer analyzed

Results

Please add representative photographs of genotypic profiles for each of RAPD-PCR primers.
Response :we added representative photographs of genotypic profiles for each of RAPD-PCR primers in supplemental materials

Please pay attention to the use of italics; genera should be written in italics (e.g. l. 128, 130, 139-149 and possibly elsewhere), species should be written with the first letter in lowercase (e.g. l. 333, 336 and possibly elsewhere)

Response :corrected in the text

Reviewer 2 Report

Comments and Suggestions for Authors

The manuscript presents well the study on the relation between origin of Brettanomyces yeast strains (mainly based on the genotype and cluster-based analysis) and  some oenological parameters like ethanol potential, volatile acidity and production of ethyl phenols.

The paper is well written and designed but it needs further revision:

line 8. before the first statement, the authors could add at least a short information about Brettanomyces here also in the abstract.In particular for non-expert readers. Is it in the family Saccharomycetaceae classified?

Is the country Italy or Italyl is part of the e-mail provided?

line 34. please check spaces here and throughout the manuscript

line 44. check spaces

Please improve the introduction. Add for example information about genotyping process. The importance of identyfing strains. Maybe Some information from the literature etc

line 64. some times Brettanomyces is written in italic, some times not.

line 64. please check spaces

line 76. please write specie names in italic

line 94 and 96. check spaces

line 222. please check the proper Celsius symbol

line 243. pleae check the paragraph and the spaces.

line 248. please check the proper celsius symbol

line 250. please check the proper symbols

line 261. please provide the authors from the citation 45

line 263. please check double paragraphs

Please review the conclusion section. It should be optimized to highlight the main findings. what did the group analysis provided?

conclusion: the authours can emphasize one of the main conclusions :" the geographical origin has a different con-159 tribution to the population structure strongly suggesting an anthro\ pic influence on the 160 spatial biodiversity of this \ microorganism. "

Author Response

The manuscript presents well the study on the relation between origin of Brettanomyces yeast strains (mainly based on the genotype and cluster-based analysis) and  some oenological parameters like ethanol potential, volatile acidity and production of ethyl phenols.

The paper is well written and designed but it needs further revision:

line 8. before the first statement, the authors could add at least a short information about Brettanomyces here also in the abstract.In particular for non-expert readers. Is it in the family Saccharomycetaceae classified?

Response :Yes, Brettanomyces yeasts are in the family Saccharomycetaceae. We added a short phrase on their importance in the fermentation industry.

Is the country Italy or Italyl is part of the e-mail provided?

Response :Corrected in the text

line 34. please check spaces here and throughout the manuscript

Response :Corrected in the text

line 44. check spaces

Response : Corrected in the text

Please improve the introduction. Add for example information about genotyping process. The importance of identyfing strains. Maybe Some information from the literature etc

line 64. some times Brettanomyces is written in italic, some times not.

Response :Corrected in the text

line 64. please check spaces

Response :Corrected in the text

line 76. please write specie names in italic

Response :Corrected in the text

line 94 and 96. check spaces

Response :Corrected in the text

line 222. please check the proper Celsius symbol

Response :Corrected in the text

line 243. pleae check the paragraph and the spaces.

Response :Corrected in the text

line 248. please check the proper celsius symbol

Response :Corrected in the text

line 250. please check the proper symbols

Response :Corrected in the text

line 261. please provide the authors from the citation 45

Response :Corrected in the text

line 263. please check double paragraphs

Response :Corrected in the text

Please review the conclusion section. It should be optimized to highlight the main findings. what did the group analysis provided?

conclusion: the authours can emphasize one of the main conclusions :" the geographical origin has a different con-159 tribution to the population structure strongly suggesting an anthro\ pic influence on the 160 spatial biodiversity of this \ microorganism. "

Response :Added the suggestion in the text